# RETHINKING LEARNING-BASED SYMMETRIC CRYPT-ANALYSIS: A THEORETICAL PERSPECTIVE

## ABSTRACT

The success of deep learning in cryptanalysis has been largely demonstrated empirically, yet it lacks a foundational theoretical framework to explain its performance. We bridge this gap by establishing a formal learning-theoretic framework for symmetric cryptanalysis. Specifically, we introduce the Coin-Tossing (CoTo) model to abstract the process of constructing distinguishers and propose a unified algebraic representation, the Conjunctive Parity Form (CPF), to capture a broad class of traditional distinguishers without needing domain-specific details. Within this framework, we prove that any concept in the CPF class is learnable in subexponential time in the setting of symmetric cryptanalysis. Guided by insights from our complexity analysis, we demonstrate preprocessing the data with a flexible output generating function can simplify the learning task for neural networks. This approach leads to a state-of-the-art practical result: the first improvement on the deep learning-based distinguisher for SPECK32/64 since 2019, where we enhance accuracy and extend the attack from 8 to a record 9 rounds.

## 1 INTRODUCTION

The rapid advancement of machine learning has significantly impacted variousfields, such as image processing and speech recognition. Recently, its success has drawn growing interest from the cryptography community, particularly in the context of symmetric cryptanalysis. As early as Rivest (1991), they envisioned the integration of machine learning with block ciphers. Although this concept was explored by researchers over the ensuing decades, it was not until Gohr (2019) developed a deep learning-based distinguisher for block cipher SPECK32/64 that outperformed traditional differential distinguishers, that neural networks began to attract significant attention from the cryptanalysis community. This breakthrough marked a turning point, bringing machine learning methods to the forefront of modern symmetric cryptanalysis. Since then, machine learning-based symmetric cryptanalysis has evolved into several key research directions, see, Gerault et al. (2024).

Currently, in cryptography community, one major line of research focuses on interpreting differential neural distinguishers. For example, Benamira et al. (2021) analyzed the information captured by each layer in the neural network and conducted experiments to verify their conjectures. They conjectured that the neural network captures specific differential conditions in the penultimate or antepenultimate rounds and replaced less interpretable components of the original network with more explainable alternatives. While Bao et al. (2023) argued that the superiority of neural differential distinguishers over classical ones is primarily due to their exploitation of certain XOR patterns. Additionally, research employing pruning techniques or visualization algorithms to improve network interpretability are discussed (Băcuieți et al., 2022).

Another significant research area aims to increase the accuracy of neural network distinguishers to achieve more effective attacks since the accuracy will greatly effect the time and data complexity during the attack. The primary approaches for improvement include optimizing the neural network algorithms and modifying the construction of the sample sets. For example, (Benamira et al., 2021) replaced ResNet blocks to maintain original accuracy while enhancing interpretability. Furthermore, they also introduced DBitNet, which is based on dilated convolutional layers, and it enables the neurons to learn the relationship between distanced bits. For ASCON permutation, Shen et al. (2024) used multilayer perceptron (MLP) to get a 4-round distinguisher. Other researchers worked on modifying the data set to improve the accuracy (Bellini et al., 2023b; Baksi et al., 2021). On the other hand,

many studies focus on enhancing the application of neural network distinguishers in key recovery attacks. Bao et al. (2022) proposed a comprehensive framework for developing key recovery attacks, introducing improved 12-round and the first 13-round key recovery attacks for SPECK32/64 , and further improved the framework in Bao et al. (2023). Some researchers have explored methods for finding better input differences through SAT solvers Hou et al. (2021), principal component analysis Seok et al. (2024) or various metrics Bellini et al. (2023b).

Despite the abundance of experimental studies on neural network distinguishers, few investigations explore the application of machine learning in block cipher cryptanalysis from a theoretical standpoint. This limitation raises several issues. First, the construction of neural distinguishers currently lacks a solid theoretical foundation, leading to a predominance of heuristic-based experiments aimed at accuracy optimization. Additionally, there is a theoretical gap concerning the feasibility of employing machine learning-based algorithms for cryptanalysis task. Moreover, the threat posed by neural networks to block ciphers remains challenging to quantify, which adversely affects both the analysis and design of block ciphers. Therefore, it is crucial to conduct research grounded in solid theoretical foundations, which can bring new perspectives for these areas.

**Contribution.** We make a dual contribution that bridges learning theory and symmetric cryptanalysis. From a theoretical perspective, we introduce a new model and a set of problems that formalize the application of deep learning in cryptanalysis. This establishes a clear and significant research objectives for the area. From a practical perspective, we leverage the conclusions from our theoretical framework to construct effective deep neural network distinguishers, thus contributing to the advancement of modern cryptanalysis.

Specifically, in this work, we first revisit the scenarios and tasks of symmetric cryptanalysis through the perspective of learning theory, highlighting the connections between cryptanalysis and ML algorithms by proposing the Coin-Tossing model. We then categorize traditional cryptanalysis methods based on Boolean functions and introduce a generalized Boolean concept grounded in conjunctive parity, namely the conjunctive parity form (CPF) concept class. By introducing the aforementioned new definitions and problems, we demonstrate the following findings:

- In the Coin-Tossing model, if a concept can be used as a distinguisher and it possesses a general CPF form, then a learning algorithm with sub-exponential complexity exists that can identify it.
- If the Hamming weight of each clause within this concept is constant, then a polynomial-complexity algorithm exists that can identify it with limited error.

Our empirical analysis further reveals that the practical complexity for various machine learning algorithms to solve this problem is strongly correlated with certain problem parameters, often showing an exponential decay. This insight motivates a novel approach: by strategically modifying the input and output generating functions, we reduce the problem's complexity upper bound, thereby boosting the efficacy of machine learning-based cryptanalysis. To validate our methodology, we applied it to the ISO-standard block cipher SPECK32/64, achieving the best-known cryptanalytic result for 8 rounds and presenting the first 9-round neural distinguisher. We further demonstrate the generalizability of our approach by improving upon existing result for DES cipher.

## 2 PRELIMINARIES

Given the interdisciplinary nature of this work, this section provides the necessary background knowledge to facilitate understanding for researchers from diverse academic fields.

### 2.1 DEEP LEARNING-BASED CRYPTANALYSIS

Block ciphers are a foundational component of modern internet communication, rendering their security of paramount importance. A block cipher is a deterministic function $E$ that maps a fixed-length plaintext $x \in \mathbb{F}_2^l$ to a ciphertext $y \in \mathbb{F}_2^l$ under a secret key $k$, denoted as $E_k(x) = y$.

A seminal technique in the chosen-plaintext attack (CPA) setting is differential cryptanalysis. An adversary selects pairs of plaintexts with a specific XOR difference such as $(x, x \oplus \Delta)$, $\Delta \in \mathbb{F}_2^l$, and analyzes the statistical properties of the corresponding cipher pair $(E_k(x), E_k(x \oplus \Delta))$.

Building upon this foundation, Gohr (2019) introduced a deep learning-based approach to cryptanalysis. The core idea is to train a neural network to differentiate between ciphertext pairs produced by a specific differential and those drawn from a random distribution. To achieve this, a balanced training dataset is constructed. Positive samples consist of valid ciphertext pairs $(E_k(x), E_k(x \oplus \Delta))$ for a fixed $\Delta$, with randomly sampled plaintext $x$ and keys $k$. Negative samples are composed of concatenated random bit strings of equivalent length. A Residual Network (ResNet) is then trained as a binary classifier on this dataset, optimized using a Mean Squared Error (MSE) loss function with L2 weight regularization.

Neural-aided cryptanalysis has emerged as a powerful technique, in some instances outperforming traditional cryptanalytic methods against certain block ciphers. As the field expands beyond differential cryptanalysis to incorporate integral and linear characteristics, a more generalized notation becomes necessary, see Hou et al. (2020); Zahednejad & Lyu (2022). This work thus adopts a universal framework and notations.

We consider a standard distinguishing scenario where an adversary has access to an encryption oracle. For $i$-th query, the adversary submits a structured plaintext vector $\boldsymbol{x^i} = (x_0, x_1, ..., x_{m-1}) \in \mathbb{F}_2^{m \times l}$. A positive sample, denoted $(\boldsymbol{y^i}, b_i = 1)$, consists of the corresponding ciphertext vector $\boldsymbol{y^i} = (\boldsymbol{y}_0, \boldsymbol{y}_1, ..., \boldsymbol{y}_{m-1}) = (E_k(x_0), E_k(x_1), ..., E_k(x_{m-1})) \in \mathbb{F}_2^{m \times l}$ represents ciphertexts encrypted under a randomly chosen key $k$. In contrast, we use $\boldsymbol{r}^i$ to represent the random sample, where $\boldsymbol{r}^i = (r_0, r_1, ..., r_{m-1})$, with each $r_t$ $(t = 0, ..., m - 1)$ being sampled in $\mathcal{U}(\mathbb{F}_2^l)$.

To construct an effective distinguisher, the adversary typically employs structured inputs designed to introduce statistical discrepancies between the distributions of positive and negative samples. Formally, each plaintext vector $\boldsymbol{x}^i$ derived from a random vector $x_i$ using a sequence of *input generating functions*: $\boldsymbol{x^i} = (\alpha_0(x_i), \alpha_1(x_i), ..., \alpha_{m-1}(x_i))$ where each $\alpha_j : \mathbb{F}_2^l \to \mathbb{F}_2^l$ and the $x_i$ is sampled uniformly. The parameter $m$ corresponds to the number of input pairs in some works. Similarly, *output generating function* $\omega$ operates on samples $\boldsymbol{y}^i$ and $\boldsymbol{r}^i$, though in most analyses it is assumed to be the identity function.

## 2.2 THE LEARNING THEORY AND LPN PROBLEM

The Probably Approximately Correct (PAC) model and framework proposed by Valiant (1984) can be described by the following definitions. Assume an adversary has access to an oracle that returns "Positive (label 1)" if $h_*(x) = 1$ and "Negative (label 0)" otherwise where the $x$ is randomly draw from a distribution $\mathcal{D}$ and the $h_*$ is the target concept. The adversary's task is to identify $h_*$ from the finite hypothesis space $\mathcal{H} = \{h_1, h_2, ..., h_N\}$. Define: $L_{\mathcal{D},f}(h) = \Pr_{x \sim \mathcal{D}}[h(x) \neq f(x)]$, then the identification procedure is said to do *probably approximately correct identification* of $h_*$ if and only if the algorithm returns a hypothesis $h$ such that, with probability of at least $1 - \delta$, $L_{\mathcal{D},h_*}(h) \leq \epsilon$ for every $\epsilon, \delta \in (0, 1)$. In our work, we abbreviate it as "pac-identification".

On the other hand, the *random classification noise model* (RCN, or simple noise model) introduced by Angluin & Laird (1988) is a specialized model within the PAC learning framework, and it makes it more general by introducing the presence of noise. Specifically, after the oracle draws an example $x$ from the distribution $\mathcal{D}$, it will return the wrong label with probability $\tau$ ($\tau < 1/2$). In addition to the standard PAC and RCN models, researchers have introduced various other frameworks, such as the SQ (Statistical Query)(Reyzin (2020); Shalev-Shwartz & Ben-David (2014)), to investigate related problems. A prominent example is the Learning Parity with Noise (LPN) problem, which is defined as follows:

**Definition 1.** *(LPN problem, Esser et al. (2017a)) In the $LPN_{n,\tau}$ problem, for a secret $\boldsymbol{s} \in \mathbb{F}_2^n$ and error parameter $\tau \in [0, \frac{1}{2})$ we are given access to an oracle that provides samples of the form:*

$$(\boldsymbol{a}_i, b_i) := (\boldsymbol{a}_i, \langle \boldsymbol{a}_i, \boldsymbol{s} \rangle \oplus e_i), \text{ for } i = 1, 2, ...$$

*where $\boldsymbol{a}_i \sim \mathcal{U}(\mathbb{F}_2^n)$ and $e_i \sim Ber_\tau$ independently. The goal is to recover $\boldsymbol{s}$.*

In other words, the oracle will draw a sample $\boldsymbol{a}$ from $\mathcal{U}(\mathbb{F}_2^n)$ and then compute the result $\theta(\boldsymbol{a}) = \langle \boldsymbol{a}, \boldsymbol{s} \rangle$ which will be corrupted by noise with probability $\tau$. Here, $\langle \cdot, \cdot \rangle$ represents the inner product of two Boolean vectors. It is noted that the LPN problem is a specific instance of the RCN model. Specifically, it assumes that the distribution $\mathcal{D}$ is a uniform distribution and that both the target Boolean function and the hypothesis space consist of parity functions. In learning theory, Feldman

et al. (2006) proved that learning DNF expressions and $k$-juntas[1] are related to the LPN problem. The interesting is that the LPN problem is not only considered in learning theory, but it also has been widely explored in cryptography, particularly in constructing cryptographic schemes based on LPN or its variants, and in efforts to reduce the complexity of solving it, see Boyle et al. (2020); Brakerski et al. (2020; 2019); Yu & Zhang (2021).

# 3 THE COIN-TOSSING MODEL

In this section, we try to explore the complexity of learning algorithms that rely solely on dataset when exploring classical analytical distinguishers. We first revisit the related neural network experiments by presenting a new model to represent the deep learning applied to a neural distinguisher in symmetric cryptanalysis. In the scenario of the CPA model, the adversary can query an encryption oracle with $m$ plaintext messages $\boldsymbol{x^i}$. Upon receiving the query, the oracle tosses a coin: if it lands heads, it returns a positive sample and its label $(\boldsymbol{y^i}, b_i = 1)$. Otherwise, it returns $(\boldsymbol{r^i}, b_i = 0)$.

By querying the oracle multiple times and then performing the function $\omega$, the adversary accumulates a dataset $\mathcal{T}$ consisting of both sample types unordered. We denote the subsets of positive and negative samples in $\mathcal{T}$ by $\mathcal{Y}$ and $\mathcal{R}$ respectively. The combined dataset $\mathcal{T}$ serves as the training dataset for the neural distinguisher. The goal of a learning-based adversary is to use a learning algorithm $\mathcal{A}$, given access to a training dataset $\mathcal{T}$, to gain an advantage in distinguishing whether an unseen vector is generated randomly from the oracle. And we denote the test dataset as $\mathcal{T}' = \{\mathcal{Y}', \mathcal{R}'\}$. A validation dataset similarly be defined for tuning hyperparameters or early stopping.[2]

Let $\Theta_\lambda = \{\theta | \theta : \mathbb{F}_2^{m \times l} \to \mathbb{F}_2\}$ denote the *concept class* parameterized by a *security parameter* $\lambda \in \mathbb{R}_{(0,1)}$ where each *concept* $\theta \in \Theta_\lambda$ satisfies the following condition:

$$\left| \mathbb{E}_{\boldsymbol{y}}[\theta(\boldsymbol{y})] - \mathbb{E}_{\boldsymbol{r}}[\theta(\boldsymbol{r})] \right| > \lambda. \tag{1}$$

Here, $\boldsymbol{y}$ and $\boldsymbol{r}$ are randomly drawn from $\mathcal{Y}$ and $\mathcal{R}$, respectively. We denote the value on the left-hand side of the inequality as $\varepsilon_\theta$, which is generally estimated from an independent test dataset. It is noted that any concept $\theta \in \Theta_\lambda$ is a distinguisher for the primitive. Then, the training data set can be regarded as following a sample distribution $\mathcal{D}_{\text{CT}}$ and the adversary needs to find one (or many) concept(s) $\hat{\theta} \in \Theta_\lambda$ consistent with $\mathcal{D}_{\text{CT}}$. In our work, we call this game-like model the *Coin-Tossing (CoTo) model*. Formally, we have the following definition.

**Definition 2.** *(The Coin-Tossing Model) Assume the adversary can query the Oracle with encryption function $E$, input generating functions $\{\alpha_i\}_{i \leq m-1}$ and output generating function $\omega$:*

$$Oracle_{CT} = \begin{cases} (\omega(\boldsymbol{y}^i), 1) = \left( \omega\big(E_{k_i}(\alpha_0(x_i)), E_{k_i}(\alpha_1(I_1(x_i))), ..., E_{k_i}(\alpha_{m-1}(x_i))\big), 1 \right) & \text{, if } b_i = 1; \\ (\omega(\boldsymbol{r}^i), 0) & \text{, otherwise.} \end{cases}$$

*The variables such that: $x_i \sim \mathcal{U}(\mathbb{F}_2^l)$, $k_i \sim \mathcal{U}(\mathbb{F}_2^\kappa)$, $\boldsymbol{r}^i \sim \mathcal{U}(\mathbb{F}_2^n)$, $b_i \sim Ber_{\frac{1}{2}}$ and $n = l \times m$. Then, the goal of adversary is return a concept $\hat{\theta} \in \Theta_\lambda$ for the given security parameter.*

This model is particularly useful because it not only encapsulates all relevant prior work at a high level of abstraction but also enables the straightforward derivation of useful insights, as demonstrated in the following remarks.

**Remark 1.** *Given that for any $\theta \in \Theta_\lambda$, the concept $\theta \oplus 1$ is also in $\Theta_\lambda$, so we can omit the constant term from our subsequent discussion.*

**Remark 2.** *Let $K_\theta := \frac{wt(\theta)}{2^{m \times l}}$ where $wt(\theta)$ is the Hamming weight of $\theta$, or equally $wt(\theta) = \left| \{x | \theta(x) = 1\} \right|$. If the adversary knows $K_\theta$, they can determine whether $\theta \in \Theta_\lambda$ based solely on the set of positive samples since the $\mathbb{E}_{\boldsymbol{r}}[\theta(\boldsymbol{r})]$ can be approximated by $K_\theta$.*

Remark 2 indicates that for a concept $\theta$ of a known form, the attack complexity can be reduced by calculating its Hamming weight. This approach is particularly useful when the algebraic properties

---

[1]Boolean functions in $\mathcal{B}_n$ that depend on at most $k$ of its variables.

[2]Without loss of generality, we assume $|\mathcal{Y}| = |\mathcal{R}| = N$ for training dataset and $|\mathcal{Y}'| = |\mathcal{R}'| = N'$ for test dataset.

of $\theta$ are known. Following the framework, we are positioned to formally address the central question: *How do machine learning-based algorithms impact the security of symmetric ciphers?*

In classical cryptanalysis, the hypothesis space is often well-defined, comprising structures such as differential or linear characteristics. In contrast, the hypothesis space explored by a neural network is substantially larger and more complex, making the learned features challenging to interpret.

# 4 THE CPF CONCEPT CLASS

Rather than directly analyzing the relationship between the hypothesis space and the concept space $\Theta_\lambda$, this section introduces the CPF concept, a concept class capable of representing many traditional cryptanalytic distinguishing methods. We first examine the differential and differential-linear distinguishers as examples.

**Example 1.** *(Differential distinguisher, (Biham & Shamir, 1991)) In a basic differential attack, a cryptanalyst identifies $(\Delta, \nabla)$ as a differential with high probability over the (round-reduced) encryption function $E_r$. Typically, in this setting, $m = 2$ and the input generating functions are $\alpha_0(x_i) = x_i$ and $\alpha_1(x_i) = x_i \oplus \Delta$, where $\Delta, \nabla \in \mathbb{F}_2^l$ denote the input and output differences, respectively. And the plaintext values $x_i$ are drawn in $\mathcal{U}(\mathbb{F}_2^l)$. Under these conditions. Let $I_\nabla$ denote the index set of bit components where $\nabla$ is 1, and let $\overline{I_\nabla}$ be its complement. Then, a distinguisher $\theta$ can be formulated as a Boolen concept: $\theta(\boldsymbol{y}) = \bigwedge_{i \in I_\nabla}(y_{0,i} \oplus y_{1,i}) \wedge \bigwedge_{i \in \overline{I_\nabla}}(y_{0,i} \oplus y_{1,i} \oplus 1)$ where $y_{0,i}$ and $y_{1,i}$ represent the $i$-th components of first and second ciphertexts, respectively. This formulation naturally extends to truncated differentials, where $I_\nabla$ represents the set of active bit indices. We define the collection of these Boolean concepts as the differential function family:*

$$\mathcal{C}_{diff}^n = \left\{ \theta : \mathbb{F}_2^n \to \mathbb{F}_2 \Big| \theta(x_1, ..., x_n) = \bigwedge_{(i,j) \in I}(x_i \oplus x_j \oplus c_{i,j}) \right\}. \tag{2}$$

The index set $I$ is a subset of $[n] \times [n]$, and for all $(i, j) \in I$, there does not exist $(i', j') \in I$ such that any two of $i, j, i', j'$ are equal. To simplify the expression, we denote $n = m \times l$ as the total input dimension. Moreover, the functions have two variables in every clause and each variable appears at most once. Next, we consider the Differential-Linear (DL) attack scenario.

**Example 2.** *(Differential-Linear distinguisher, (Langford & Hellman, 1994; Beierle et al., 2022)) In the attack of the Differential-Linear method, the cryptanalyst will find a differential pair $(\Delta, \nabla)$ and a pair of linear masks $(\Gamma_{in}, \Gamma_{out})$ make the following correlation as high as possible.*

$$\left| \mathbb{E}_{x \sim \mathcal{U}(\mathbb{F}_2^n)}[\langle \Gamma_{out}, E(x) \rangle \oplus \langle \Gamma_{out}, E(x \oplus \Delta) \rangle] - \frac{1}{2} \right| = p.$$

*Similar to differential attacks, we can use a characteristic function to represent correlations as: $\theta(\boldsymbol{y}) = \bigoplus_{i \in I_\gamma}(y_{0,i} \oplus y_{1,i}) \oplus c$ where $I_\gamma$ is the index set of $\Gamma_{out}$. The corresponding input generating functions are $\alpha_0(x_i) = x_i$ and $\alpha_1(x_i) = x_i \oplus \Delta$. We employ a more general set of functions, referred to as the parity function family [3], to represent these characteristic functions:*

$$\mathcal{C}_{parity}^n = \left\{ \theta : \mathbb{F}_2^n \to \mathbb{F}_2 \Big| \theta(x_1, ..., x_n) = \bigoplus_{i \in I} x_i \oplus c, I \subseteq [n] \right\}. \tag{3}$$

Notice that the function family in equation 2 and 3 can be unified under a broader function family:

$$\mathcal{C}_{\text{CPF}}^{k,n} = \left\{ \theta : \mathbb{F}_2^n \to \mathbb{F}_2 \Big| \theta(x_1, ..., x_n) = \bigwedge_{j=0}^{k-1} \Big( \bigoplus_{i \in I_j} x_i \oplus c_j \Big), I_j \subseteq [n] \right\}. \tag{4}$$

Here, the $I_j$ are index sets and $c_j$ are constant variables. We refer to a Boolean function of this form as a *Conjunctive Parity Form (CPF) function*. In geometry, the satisfying assignments of these functions can be interpreted as the intersection of a collection of affine hyperplanes in $\mathbb{F}_2^n$, each defined by a specific index set $I_j$ and constant $c_j$.

In addition to the two examples mentioned above, many other distinguishers, such as those based on multiple differential, linear, impossible differential, integral, and boomerang distinguishers, can also be represented using this framework. The expressions of these distinguishers in the CPF Boolean function family are summarized in the following table, where the column "CL" is the variable number in each clause (or Hamming weight), and "CN" refers to the clause number $k$.

---

[3]Truth table of these functions contain an equal number of zeros and ones, namely, they are balanced.

| Distinguisher Type | Input Generating Functions | CL | CN | Ref. |
|---|---|---|---|---|
| (Impossible) differential | $(E(x), E(x \oplus \Delta))$ | 2 | $l$ | (Knudsen, 1998) |
| Rotation differential | $(E(x), E((x \lll t) \oplus \Delta))$ | 2 | $l$ | (Khovratovich & Nikolić, 2010) |
| Truncated differential | $(E(x), E(x \oplus \Delta))$ | 2 | $< l$ | (Knudsen, 1994) |
| Boomerang/Rectangle | $(E(x), E(x \oplus \Delta), E(x'),$ | 2 | $2l$ | (Wagner, 1999) |
| | $E(\Delta \oplus x'))$ | | | (Biham et al., 2001) |
| Linear | $(x, E(x))$ | - | 1 | (Matsui, 1993) |
| Differential-Linear | $(E(x), E(x \oplus \Delta))$ | - | 1 | (Langford & Hellman, 1994) |
| Integral | $\mathbb{Y}$ (output multi-set) | $|\mathbb{Y}|$ | - | (Knudsen & Wagner, 2002) |

Table 1: CPF-based distinguishers and their parameters

# 5 LEARNING A CPF CONCEPT IN COIN-TOSSING MODEL

After exploring the important concept class which can be potentially learned by adversaries, it is natural to introduce the learning theory to discuss the difficulty an adversary has in acquiring them. The Example 3 in DL distinguisher shows how the cryptanalysis is related to the LPN problem.

**Example 3.** *Recall that in the CoTo model, the oracle will toss a coin to determine whether a positive sample (generated by encryption) or a negative sample (created randomly) is returned. Suppose there exists a concept function $c$ in $\mathcal{C}_{parity}^n$ such that $\mathbb{E}_{\boldsymbol{y} \sim \mathcal{Y}}[c(\boldsymbol{y})] = p, (1/2 < p \leq 1)$, namely there exists a DL distinguisher with high probability $p$ for encrypted output. Then:*

- *In the positive sample set, $N \cdot p$ samples satisfy $c$ while $N \cdot (1-p)$ samples do not.*
- *In the negative sample set, the number of samples satisfying $c$ is $N/2$ (due to uniform randomness and the parity function is balanced).*

*Thus, among the total $(N/2 + Np)$ samples where $c(x) = 1$, exactly $N/2$ have incorrect labels—i.e., "corrupted" by noise. Similarly, there are $N(1-p)$ samples are "corrupted" by noise in $N(1-p) + N/2$ samples which do not satisfy $c$.*

Specifically, we have demonstrated that the task of searching for a DL characteristic can potentially be reduced to the LPN problem. For the more general case, we have the following Lemma 1.

**Lemma 1.** *In the CoTo model, suppose there exists a Boolean concept $\theta \in \mathcal{B}_n$ with discrepancy $\varepsilon_\theta$. Then, the probability $\tau$ of wrong label in $\theta(x) = 0$ and probability $\bar{\tau}$ of wrong label in $\theta(x) = 1$ for $x \sim \mathcal{D}_{CT}$ are given by:*

$$\tau = \Pr_{x \sim \mathcal{D}_{CT}}\left[b = 1 \big| \theta(x) = 0\right] = \frac{1 - K_\theta - \varepsilon_\theta}{2 - 2K_\theta - \varepsilon_\theta}, \quad \bar{\tau} = \Pr_{x \sim \mathcal{D}_{CT}}\left[b = 0 \big| \theta(x) = 1\right] = \frac{K_\theta}{2K_\theta + \varepsilon_\theta},$$

*where the $K_\theta$ is $wt(\theta)/2^n$, $b$ represents the label of corresponding sample and $\mathcal{D}_{CT}$ denotes the sample distribution within the model.*

The aforementioned method and the lemma provide a basic thought for transforming the CoTo model into an RCN model. We will consistently apply this approach in the subsequent sections. Additionally, Appendix A.1 includes a diagram that visually demonstrates this transformation.

However, there are two key differences between the standard LPN problem and the above example. First, the noise does not follow a Bernoulli distribution but instead follows a conditional distribution based on the samples and the samples' distribution. Second, the oracle draws the samples with the distribution $\mathcal{D}_{\text{CT}}$ and not the uniform random distribution $\mathcal{U}(\mathbb{F}_2^n)$. So, it is natural to identify a new problem, and if an algorithm can handle LPN problems, then it needs to satisfy some specific properties if it is expected to solve the new problem.

**Definition 3.** *(LPN-M problem) Let $\mathcal{D}$ be a distribution over $\mathbb{F}_2^n$. The LPN-M$_{n,\tau,\bar{\tau}}^{\mathcal{D}}$ problem, a modified version of LPN$_{n,\tau}$ problem, is defined as follows: Given access to samples $(\boldsymbol{x}, b)$ where:*

- $\boldsymbol{x} \sim \mathcal{D}$,
- *$b$ is the label of $\langle \boldsymbol{x}, s \rangle$.*
- *If $\langle \boldsymbol{x}, s \rangle = 0$, the label is corrupted with probability $\tau$.*
- *If $\langle \boldsymbol{x}, s \rangle = 1$, the label is corrupted with probability $\bar{\tau}$.*

*And the adversary's goal is to recover the secret $s$.*

It is important to note that there has been little progress in reducing the LPN problem in recent years. In this problem, since the distribution in CoTo model not only directly determines the sample (or

matrix) distribution in the LPN problem, but is also directly coupled with the noise through Lemma 1, we believe that reducing the LPN problem to this problem is absolutely non-trivial. However, what we can try is to use the LPN solution algorithm instance to construct an exact algorithm for solving new problems, which can directly give an upper bound on the complexity of the problem.

The BKW algorithm, introduced by Blum et al. (2003), is presently the most important method for solving the LPN problem, providing a practical algorithm with sub-exponential complexity. Specifically, the example and computation-time complexity for BKW algorithm in the standard LPN problem are $\text{poly}((1-2\tau)^{-2^a}, 2^b)$ where $a \cdot b = n$. If we plugin $a = \lg n/2$ and $b = 2n/\lg n$, then it can be estimated as $2^{\mathcal{O}(n/\log n)}$. We now introduce two properties concerning this algorithm.

**Definition 4.** *(distribution-free) An algorithm is called distribution-free if it can solve $LPN_{n,\tau}$ across any sample distribution in CoTo model, within the same complexity class.*

**Definition 5.** *(well-ordered) An algorithm is well-ordered if it can solve $LPN\text{-}M_{n,\tau,\tau}^{\mathcal{D}}$ can also solve $LPN\text{-}M_{n,t_0,t_1}^{\mathcal{D}}$ where $t_0, t_1 \leq \tau$ without an increase in time complexity.*

Notably, the BKW algorithm possesses both of these properties, which enables its use as a subroutine to solve different problems. The properties of the BKW algorithm are discussed in detail in Appendix A.5. Consequently, we can establish the following theorem.

**Theorem 1.** *In a CoTo model, suppose there exists a concept function $c \in \mathcal{C}_{parity}$ with discrepancy $\varepsilon_c$. If an algorithm $\mathcal{A}$ can solve $LPN_{n,\frac{1-2\varepsilon_c}{2-2\varepsilon_c}}$ and is distribution-free and well-ordered simultaneously, then $\mathcal{A}$ can recover $c$ with the same complexity class under the CoTo model.*

**Corollary 1.** *Assume $c \in \mathcal{C}_{parity}$ in the CoTo model with a constant discrepancy $\varepsilon_c$. Then, there exists an algorithm $\mathcal{A}$ that can output $c$ with example, time and memory complexities $2^{\mathcal{O}(n/\log(n))}$ where $n$ is the bit length.*

The corollary is derived by combining the BKW algorithm and Theorem 1. These results illustrate the relationship between the LPN problem and existing cryptanalysis challenges. Consequently, we are motivated to explore the problem of learning the CPF family through the lens of the LPN problem. The benefit of this transformation is that it enables the use of existing algorithms to analyze new models. Building on this foundation, we introduce the following problem:

**Definition 6.** *(LCPN problem) In the $LCPN_{n,k,\tau}$ problem, given $k$ linearly independent secrets $\boldsymbol{s}_j \in \mathbb{F}_2^n, j \in [k]$ and a noise parameter $\tau \in [0, \frac{1}{2})$, we have access to an oracle that provides samples of the form:*

$$(\boldsymbol{a}_i, b_i) := (\boldsymbol{a}_i, \bigwedge_{j=0}^{k-1} \langle \boldsymbol{a}_i, \boldsymbol{s}_j \rangle \oplus e_i), \text{ for } i = 0, 1, ...$$

*where $\boldsymbol{a}_i \sim \mathcal{U}(\mathbb{F}_2^n)$ and $e_i \sim Ber_\tau$ independently. Our goal is to recover $k$ secrets $\boldsymbol{s}_j$.*

In the absence of noise, the problem can be easily solved by finding the solution space of $Ax = 0$, where $A = (s_1, ..., s_k)^T$ is a $k \times n$ matrix by viewing the $s_j$ as vectors. However, the introduction of noise makes the problem significantly more challenging. Specifically, when $k = 1$, then the problem reduces to the LPN problem, indicating that the worst-case hardness of LCPN is at least as great as LPN. Nevertheless, the average hardness of the question needs further study. Additionally, if the Hamming weight is restricted to a constant number independent of $n$, the LCPN problem is PAC-learnable in such cases. This conclusion can be proven with the following theorems.

**Theorem 2.** *(Angluin & Laird, 1988) In RCN model with noise rate $\tau$, for every $c \in CNF(n, h)$, there exists a learning algorithm pac-identifies $c$ with time and data complexity $\text{poly}(n^h, 1/\epsilon, \log 1/\delta, 1/(1-2\tau_b))$ where $\tau_b$ is a constant s.t. $0 < \tau \leq \tau_b < 1/2$.*

The class $CNF(n, h)$ denotes Conjunctive Normal Form (CNF) formulas over $n$ variables, with at most $h$ literals in each clause. The algorithm can be constructed since the cardinality of all possible clauses in $CNF(n, h)$ is at most $(2n + 1)^h$. When $h$ is a constant independent of $n$, the complexity remains polynomial. Additionally, determining whether a clause exists in a formula can be verified through the relevant probability calculations. Theorem 2 holds for any data distribution $\mathcal{D}$ under the RCN model. In other words, it can be readily extended similarly to the CoTo model. Furthermore, each instance in $\mathcal{C}_{CPF}$ with constant clause length can be expressed as a formula in $CNF(n, h)$. This leads to one of our main results.

**Theorem 3.** *In the CoTo model, assume there exists a concept $\theta_* \in \mathcal{C}_{CPF}^{k,n}$ with discrepancy $\varepsilon_\theta$ such that the number of variables in each clause is at most $h$ and $\theta_* \in \Theta_\lambda$. Then there exists a learning algorithm that returns $\hat{\theta} \in \Theta_\lambda$ with time complexity $poly(n^h, 1/\epsilon', \log 1/\delta, 1/(1 - 2\tau'_b))$ and probability $1 - \delta$, provided that: $\epsilon' < \frac{\varepsilon_{\theta_*} - \lambda}{2(1 - 2^{1-k})}$ and $\max(\frac{1}{2 + 2^k \varepsilon_{\theta_*}}, \frac{1 - 2^{-k} - \varepsilon_{\theta_*}}{2 - 2^{1-k} - \varepsilon_{\theta_*}}) \leq \tau'_b < 1/2$.*

This case encompasses many attacks in symmetric cryptanalysis, as illustrated in Table 1, e.g., the differential, boomerang, and rectangle attack (where the CL is constant). Consequently, an adversary could design an efficient learning algorithm to capture these concepts in this case. There are many evidences to confirm this, see pure differential distinguishers in Gohr (2019). However, in more general cases, the problem becomes intractable, and it can not be solved using the method in Theorem 2 due to the exponential cardinality of clauses. As previously mentioned, there is substantial evidence that the LPN problem is difficult to solve. Since the LPN problem is a specific instance of LCPN, it is reasonable to conjecture that the average LCPN problem cannot be solved efficiently. The following lemma show how to solve the LCPN problem using a LPN algorithm.

**Lemma 2.** *If a well-ordered and distribution-free algorithm $\mathcal{A}$ can solve the $LPN_{n,\tau'}$ problem, then there exists an algorithm that can solve $LCPN_{n,k,\tau}$ within the same complexity class, where $\tau' = \frac{1 - p - p\tau}{2 - p - 2p\tau}$ and $p = \frac{2^{k-1}}{2^k - 1}$.*

In the symmetric scenario, CPF concepts are always $\mu$-expressions, meaning each variable appears at most once. In this case, the algorithm in Lemma 2 can be slightly accelerated, the proof and details can be found in Appendix 3.

The last thing to apply Lemma 2 into symmetric cryptanalysis is to consider replacing the random uniform distribution from the sample distribution $\mathcal{D}_{CT}$ generated by CoTo model. We also introduce the LCPN-M problem like Definition 3 similarly. However, the proposed approach reduces a single LCPN instance to several independent LPN instances and solves each of them separately using the BKW algorithm. Although the underlying distribution changes at this stage, as established in Theorem 1, the BKW algorithm remains fully applicable under the coin-tossing distribution without affecting the overall complexity. Therefore, modifying the distribution of the LCPN problem to $\mathcal{D}_{CT}$ in this reduction introduces no additional issues.

Assume the discrepancy of one concept $c \in \mathcal{C}_{CPF}^{k,n}$ is $\varepsilon_c$, then it holds that $\tau = \frac{2^k - 1 - 2^k \cdot \varepsilon_c}{2^{k+1} - 2 - 2^k \cdot \varepsilon_c}$ and $\overline{\tau} = \frac{1}{2^k \cdot \varepsilon_c + 2}$. So we take the $\tau' = \max(\tau, \overline{\tau})$ in the theorem and corollary to estimate the complexity. Again, if we consider the noise rate as a variable instead of a constant, then the BKW algorithm can solve the LPN problem with time, data, and memory complexity $poly((1 - 2\tau)^{-2^a}, 2^b)$ where $n = a \cdot b$. When combined with the result from Lemma 2, the following theorem is obtained.

**Theorem 4.** *In CoTo model, assume there exists a concept $\theta \in \mathcal{C}_{CPF}^{k,n}$ with discrepancy $\varepsilon_\theta$. Then there exists a learning algorithm that returns the $\theta$ with time and space complexity $poly(k, (\frac{p}{2 - p - 2p\tau})^{2^a}, 2^b)$ where $ab \geq n$, $\tau = \max(\frac{2^k - 1 - 2^k \cdot \varepsilon_c}{2^{k+1} - 2 - 2^k \cdot \varepsilon_c}, \frac{1}{2^k \cdot \varepsilon_c + 2})$ and $p = \frac{2^{k-1}}{2^k - 1}$. If $k, \varepsilon_c$ are constants, then the complexity is $2^{\mathcal{O}(n/\log n)}$.*

The key difference between Theorem 3 and Theorem 4 lies in their assumptions. Theorem 3 assumes prior knowledge of the constant Hamming weight of each clause in the target concept, whereas Theorem 4 does not, making it a more general result. Compared with brute-force search for secrets $s_j$, which would require enumerating all possible functions within the target Boolean family and verifying each candidate (the complexity is $2^{\mathcal{O}(n)}$), the theorem provides a non-trivial solution.

Although a theoretical upper bound is provided, the complexity of learning CPF concepts in the CoTo model with DNNs and other machine learning models is yet to be determined. We therefore conduct a series of experiments to quantify the influence of crucial CPF parameters on the performance of several ML algorithms. The results show that, under a fixed noise rate, the underlying function of complexity decays exponentially with the variables $n$ and $h$. Further details on our methodology and results are provided in Appendix B.1.

| Nr | Nerual Distinguisher | Input Difference | Accuracy | TPR | TNR |
|----|----------------------|------------------|----------|-----|-----|
| 8 | Gohr (2019) | (0x0040, 0x0000) | 0.514 | 0.519 | 0.508 |
| 8 | Bellini et al. (2023b) | (0x0040, 0x0000) | 0.514 | - | - |
| 8 | Bao et al. (2023) | (0x0040, 0x0000) | 0.5135 | 0.5184 | 0.5085 |
| 8 | Traditional (Bellini et al. (2023a)) | (0x2800, 0x0010) | 0.5048 | 0.5095 | 0.5000 |
| 8 | Adaptive Gohr (2019) | (0x2800, 0x0010) | 0.5007 | 0.2774 | 0.7236 |
| 8 | This work ($\mathcal{ND}_{\text{comp}}^1$) | (0x2800, 0x0010) | **0.5205** | 0.401 | 0.639 |
| 9 | Adaptive Gohr (2019) | (0xa810, 0x0010) | 0.5000 | - | - |
| 9 | Adaptive Gohr (2019) | (0x0040, 0x0000) | 0.5000 | - | - |
| 9 | This work ($\mathcal{ND}_{\text{comp}}^2$) | (0xa810, 0x0010) | **0.5016** | 0.5031 | 0.5000 |

Table 2: The comparison of accuracy, true positive rate, and true negative rate between different distinguishers.

## 6 APPLICATIONS: ADVANCED NEURAL DISTINGUISHERS

In this section, we will focus on using the observations and conclusions in Section 5 to construct new neural distinguishers on round-reduced SPECK32/64, a ISO-standard cipher from the SPECK (Beaulieu et al., 2013). It is worth noting that SPECK32/64 is among the few block ciphers where neural network-based distinguishers have been shown to outperform classical cryptanalytic techniques, making it an ideal candidate for our study.

A crucial step in constructing a neural distinguisher is the selection of suitable input generating functions to ensure that concepts with high discrepancy exist within the concept set $\Theta_\lambda$. This is a prerequisite for a deep learning-based algorithm to capture such concepts. To this end, we first employ methods based on Mixed-Integer Linear Programming (MILP)/ Mixed-Integer Quadratically Constrained Programming (MIQCP) introduced by Benamira et al. (2021) to identify an appropriate differential and subsequently generate the corresponding input generating functions. Furthermore, guided by the complexity upper bound presented in Section 5 and the characteristics of the CPF concept class, we introduce the compression function $\omega(\boldsymbol{y}_0, \boldsymbol{y}_1) := \boldsymbol{y}_0 \oplus \boldsymbol{y}_1$ as our output generating function. This choice is designed to preserves the structural characteristics of the class while simultaneously lowering the complexity upper bound for the learning problem. Specifically, this type of function, which is independent of cryptographic structures, can reduce the upper bound on the complexity presented in Theorem 4 from $2^{\mathcal{O}(n/\log n)}$ to $2^{\mathcal{O}(n/2 \cdot \log(n/2))}$. Although this data compression narrows the concept space, it does not affect mainstream distinguishers. Table 2 presents the accuracy of our constructed neural distinguisher and provides a comparison with previous results.

Beyond the improvements to the differential neural distinguisher for SPECK32/64, we extended our method to the DES algorithm and integral-based distinguishers by setting $\omega(\boldsymbol{y}_0, ..., \boldsymbol{y}_m) = (\pi_{c_0}(\boldsymbol{y}_0), ..., \pi_{c_1}(\boldsymbol{y}_i))$, where $\pi_{c_k}$ are projections map from $\mathbb{F}_2^n$ to $\mathbb{F}_2^{hw(c_k)}$. It will reduce the complexity bound from $2^{\mathcal{O}(n/\log n)}$ to $2^{\mathcal{O}(hw(c)/\log hw(c))}$. This extension yielded two key results: an increase in the accuracy of the DES linear neural distinguisher to 0.7 from 0.668, and the direct learning of a 7-round integral neural distinguisher for SPECK32/64. These results further demonstrate the generality of our conclusions and methods. For comprehensive details on cipher structure, training pipeline, the selection of generating functions and benchmarks, see the Appendix B.2.

## 7 CONCLUSION

In this paper, we first establish a theoretical framework, the Coin-Tossing model, which serves to generalize all existing neural network-based distinguishing attacks. Concurrently, we introduce the CPF concept family to unify the algebraic expressions of classical distinguishing attacks. Within this framework, we then devise a slightly sub-exponential time algorithm for learning general CPF concepts. Motivated by this complexity analysis, we investigate the use of compression to reduce the learning complexity of neural networks under the CoTo model. In the experiment, we got the advanced deep learning-based distinguishers. A limitation of the present study is that we only establish a general upper bound on the computational complexity of the problem. We defer a more detailed investigation into the complexity of this problem to future research. Nevertheless, the underlying principles of our application can be extended to inform the development of more intricate output generating functions that leverage additional prior knowledge.

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

# Appendix

In Appendix A, we provide the detailed proofs for all lemmas and theorems presented in the main paper. The following Appendix B elaborates on the experiments conducted to investigate the learning of the CPF concept with a practical ML model and building the relevant neural distinguishers. The second part presents a discussion on the properties of the BKW algorithm. Finally, the Appendix C clarifies the usage of large language models in this work.

## A PROOFS OF THEOREMS AND LEMMAS

### A.1 PROOF OF LEMMA 1

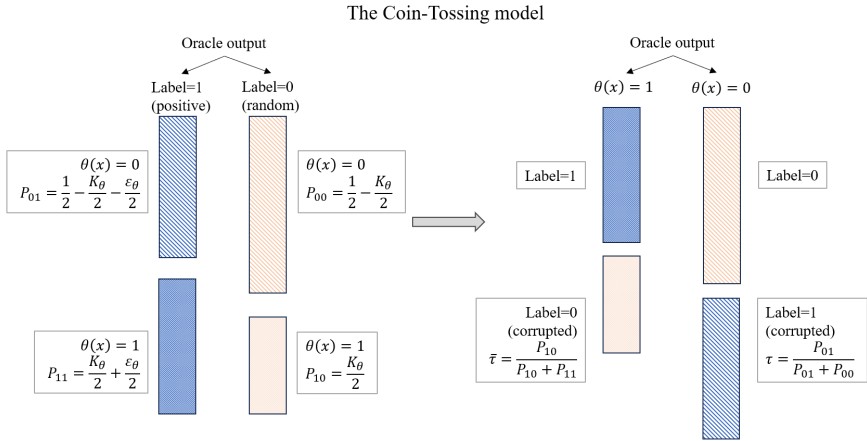

Figure 1: An alternative representation of the Coin-Tossing model, along with a graphical depiction of its relevant parameters.

Suppose that $\theta$ is a Boolean concept functioning as a distinguisher within the Coin-Tossing model. For a given sample, the relationship between the label returned by the oracle and the classification distinguisher by $\theta$ is depicted in the left portion of the Figure 1. Subsequently, this relational diagram can be reshaped as illustrated in the right half of the figure by plotting the categories determined by $\theta$ on the left and right sides, respectively. In the right figure, for the column where $\theta(x) = 0$, the instances where corresponding label is 1 (represented by the darker section) can be interpreted as noise introduced by the oracle to the sample. Similarly, in the column where $\theta(x) = 1$, the instances with corresponding label is 0 (depicted by the lighter section) correspond to outcomes corrupted by noise. Finally, drawing an analogy between this model and the LPN problem is straightforward; it suffices to consider the inner product in the LPN framework as the Boolean function $\theta$.

Next, we denote $P_{ij} := \Pr_{x \sim \mathcal{D}_{\mathcal{CT}}}[\theta(x) = i, b = j]$ where $i, j \in \{0, 1\}$. According to the definition of discrepancy, we have $|\mathbb{E}_{\boldsymbol{y}}[\theta(\boldsymbol{y})] - K_\theta| = \varepsilon_\theta$. Without loss of generality, we assume $\mathbb{E}_{\boldsymbol{y}}[\theta(\boldsymbol{y})] = K_\theta + \varepsilon_\theta = \Pr_{x \sim \mathcal{D}_{\mathcal{CT}}}[\theta = 1|b = 1]$. Here, we use the $K_\theta$ to approximate $\mathbb{E}_{\boldsymbol{r}}[\theta(\boldsymbol{r})]$, and it equals to $\Pr_{x \sim \mathcal{D}_{\mathcal{CT}}}[\theta = 1|b = 0]$. On the other hand, $\Pr[b = 0] = \Pr[b = 1] = 1/2$ in the Coin-Tossing model. We therefore can get the values of $P_{ij}$ as showing in the figure. Then the conditional noise rate $\overline{\tau}$ and $\tau$ in the Lemma 1 can be calculated as:

$$\overline{\tau} = \frac{P_{10}}{P_{10} + P_{11}} = \frac{K_\theta}{2K_\theta + \varepsilon_\theta}, \quad \tau = \frac{P_{01}}{P_{01} + P_{00}} = \frac{1 - K_\theta - \varepsilon_\theta}{2 - 2K_\theta - \varepsilon_\theta}.$$

If the $\theta$ has the form of CPF, then $K_\theta = \frac{1}{2^k}$ (assume the clauses in $\theta$ are independent and different). Then the corresponding noise rate can be easily calculated through the conditional probability.

### A.2 PROOF OF THEOREM 1

By setting $K_c = \frac{1}{2}$ into Lemma 1, the condition noise become $\tau = \frac{1-2\varepsilon_c}{2-2\varepsilon_c}$ and $\overline{\tau} = \frac{1}{2+2\varepsilon_c}$, with $\tau \geq \overline{\tau}$ (since $\varepsilon_c + K_c \leq 1$). According to the definition of well-ordered, we take the maximum

value of $\tau$ and $\overline{\tau}$, denoted as $\tau'$. This implies that the algorithm $\mathcal{A}$ capable of solving LPN-M$_{n,\tau',\tau'}^{\mathcal{D}}$, can also solve LPN-M$_{n,\tau,\overline{\tau}}^{\mathcal{D}}$ within the same complexity class.

## A.3 PROOF OF THEOREM 3

According to Lemma 1, substituting $1/2^k$ into $K_{\theta_*}$ yields the two noise rate conditions as follows:

$$\tau = \frac{1 - K_{\theta_*} - \varepsilon_{\theta_*}}{2 - 2K_{\theta_*} - \varepsilon_{\theta_*}} = \frac{1 - 2^{-k} - \varepsilon_{\theta_*}}{2 - 2^{1-k} - \varepsilon_{\theta_*}},$$

$$\overline{\tau} = \frac{K_{\theta_*}}{2K_{\theta_*} + \varepsilon_{\theta_*}} = \frac{2^{-k}}{2^{1-k} + 2\varepsilon_{\theta_*}} = \frac{1}{2 + 2^k \varepsilon_{\theta_*}}.$$

And the $\tau_b'$ is the upper bound of the noise rate, so it is simply greater than or equal to $\max(\frac{1}{2+2^k \varepsilon_{\theta_*}}, \frac{1-2^{-k}-\varepsilon_{\theta_*}}{2-2^{1-k}-\varepsilon_{\theta_*}})$. Then we will consider the case that after introducing the error $\epsilon$, the algorithm can still return the concept $\theta_* \in \Theta_\lambda$.

Let $\Pr_{x \sim \mathcal{D}_{\mathrm{CT}}}[\hat{\theta}(x) \neq \theta_*(x)] = \epsilon'$, $\Pr_{x \sim \mathcal{Y}}[\hat{\theta}(x) \neq \theta_*(x)] = \epsilon_1$, and $\Pr_{x \sim \mathcal{R}}[\hat{\theta}(x) \neq \theta_*(x)] = \epsilon_2$. According to the definition of discrepancy of concept $\hat{\theta}$, it holds that:

$$\varepsilon_{\hat{\theta}} = |\mathbb{E}_{x \sim \mathcal{Y}}[\hat{\theta}(x)] - \mathbb{E}_{x \sim \mathcal{R}}[\hat{\theta}(x)]|.$$

Without loss of generality, we assume that the value inside the absolute value is greater than zero. Then it can be calculated as:

$$\varepsilon_{\hat{\theta}} = (\frac{1}{2^k} + \varepsilon_{\theta_*})(1 - \epsilon_1) + (1 - \frac{1}{2^k} - \varepsilon_{\theta_*})\epsilon_1 - \frac{1}{2^k}(1 - \epsilon_2) - (1 - \frac{1}{2^k})\epsilon_2$$

$$= 2\epsilon_1(1 - \frac{1}{2^{k-1}} - \varepsilon_{\theta_*}) - 2\epsilon'(1 - \frac{1}{2^{k-1}}) + \varepsilon_{\theta_*}$$

$$\geq \varepsilon_{\theta_*} - 2\epsilon'(1 - 2^{1-k}) \geq \lambda$$

$$\Rightarrow \epsilon' \leq \frac{\varepsilon_{\theta_*} - \lambda}{2(1 - 2^{1-k})}.$$

The second equality holds due to $\epsilon_1 + \epsilon_2 = 2\epsilon'$. The subsequent inequality follows by setting $\epsilon_1 = 0$, given the fact that $1 - 2^{1-k} - \varepsilon_{\theta_*} > 0$. Finally, the conclusion is derived from the definition $\varepsilon_{\hat{\theta}} \geq \lambda$ implies $\hat{\theta} \in \Theta_\lambda$.

## A.4 PROOF OF LEMMA 2 AND ITS APPLICATION

First, we need to introduce the following lemma:

**Lemma 3.** *Let $x_0, x_1, ..., x_{k-1}$ be i.i.d variables following $Ber_{1/2}$. Then for any $s \in [k]$, $\Pr[x_s = 1 | \prod_{i=0}^{k-1} x_i = 0] = \frac{2^{k-1}-1}{2^k-1}$ and $\Pr[x_s = 0 | \prod_{i=0}^{k-1} x_i = 0] = \frac{2^{k-1}}{2^k-1}$.*

*Proof.*

$$\Pr[x_s = 1 | \prod_{i=0}^{k-1} x_i = 0] = \frac{\Pr[x_s = 1, \prod_{i=0, i \neq s}^{k-1} x_i = 0]}{\Pr[\prod_{i=0}^{k-1} x_i = 0]} = \frac{(1 - 1/2^{k-1})/2}{1 - 1/2^k}.$$

$\square$

According to Lemma 3, assume that $\Pr[\langle a_i, s_t \rangle = 0 | \bigwedge_{j=0}^{k-1} \langle a_i, s_j \rangle = 0] = \frac{2^{k-1}}{2^k-1} = p$. Since $\Pr[\langle a_i, s_t \rangle = 0 | \bigwedge_{j=0}^{k-1} \langle a_i, s_j \rangle = 1] = 0$, it is easy to verify:

$$\Pr\left[\langle a_i, s_t \rangle = 0 \Big| \bigwedge_{j=0}^{k-1} \langle a_i, s_j \rangle \oplus e = 1\right] = \tau \cdot p,$$

$$\Pr\left[\langle a_i, s_t \rangle = 1 \Big| \bigwedge_{j=0}^{k-1} \langle a_i, s_j \rangle \oplus e = 0\right] = 1 - p - p \cdot \tau.$$

Then the problem can be regarded as the noise appears in negative samples with probability $\frac{\tau}{1+2\tau}$, and in positive samples with probability $\tau' = \frac{1-p-p\tau}{2-p-2p\tau}$. Furthermore, it holds that $\frac{\tau}{(1+2\tau)} < \frac{(1-p-p\tau)}{(2-p-2p\tau)} < 1/2$ for $0 < \tau < 1/2$, thus they are valid noise parameters.

Given this, a well-ordered algorithm $A$ capable of solving $\text{LPN}_{n,\tau'}$ can also solve a certain secret from this problem. Without loss of generality, we denote the first solved secret as $s_1'$. Now consider any vector $w$ that is independent of all $s_i$ ($i = 1, ..., k$), it holds that: $\Pr[\langle a_i, w \rangle = 0] = \Pr[\langle a_i, w \rangle = 1] = 1/2$ and these probabilities are independent of $\Pr[\bigwedge_{j=0}^{k-1} \langle a_i, s_j \rangle \oplus e]$. It means that we can query the oracle and apply a filter to make the samples such that:

$$\Pr\left[\langle a_i, s_1' \rangle = 0 \Big| \bigwedge_{j=0}^{k-1} \langle a_i, s_j \rangle \oplus e = 1\right]$$

$$= \Pr\left[\langle a_i, s_1' \rangle = 1 \Big| \bigwedge_{j=0}^{k-1} \langle a_i, s_j \rangle \oplus e = 0\right] = \frac{1}{2}.$$

This filter breaks the correlation between the label and $s_1$ without affecting the other secrets, as they remain independent. If the algorithm $\mathcal{A}$ needs $D$ samples, we can query extra $\frac{D}{(1-\tau)}$ samples to ensure that there are $D$ samples irrelevant to secret $s_1'$. Additionally, if there are $a$ samples with a positive label among $N$ total samples, then $\tau$ can be estimated empirically as:

$$\frac{a}{N} = \Pr\left[\bigwedge_{j=0}^{k-1} \langle a, s_j \rangle = 0, e = 1\right] + \Pr\left[\bigwedge_{j=0}^{k-1} \langle a, s_j \rangle = 1, e = 0\right]$$

$$\Rightarrow \tau = \left(\frac{a}{N} - \frac{1}{2^k}\right) \cdot \left(\frac{2^{k-1}}{2^{k-1}-1}\right).$$

Repeating this procedure $k$ times enables recovery of all secrets $s_j$, and since $k$ is independent of $n$, the overall complexity remains in the same class. Next we can apply the lemma into the symmetric cryptanalysis by the following corollary:

**Corollary 2.** *If an algorithm $\mathcal{A}$ can solve the $\text{LPN}_{n,\tau'}$ problem with space complexity $S_{n,\tau'}$ and time complexity $T_{n,\tau'}$, then for a specific $\text{LCPN}_{n,k,\tau}$ problem where the support sets of secrets are pairwise disjoint, it can be solved with time complexity of $k(T_{n,\tau'} + S_{n,\tau'})$ and space complexity of $S_{n,\tau'}$ where $\tau'$ remains consistent with Lemma 2.*

*Proof.* The algorithm can be slightly modified based on the proof of Lemma 2. During the sieving phase, additional data is unnecessary because the algorithm can simply remove the columns corresponding to the non-zero entries in the secret. The cumulative time complexity over $k$ rounds is bounded by

$$\sum_{i=1}^{k} (T_{n-\sum_{j=1}^{i} h_j, \tau'} + S_{n-\sum_{j=1}^{i} h_j, \tau'}) \leq k(T_{n,\tau'} + S_{n,\tau'}).$$

The space complexity remains unchanged. $\qquad\square$

### A.5 THE PROPERTIES OF BKW ALGORITHM

The BKW algorithm is introduced by Blum, Kalai and Wasserman to solve LPN problem which is based on block Gaussian elimination Blum et al. (2003). We adopt a high-level description of BKW in Esser et al. (2017b) to illustrate the algorithm. The algorithm in 1 is referred to one iteration to compute the first bit of secret $s$, and the other bits analogous. In this section, we will discuss why the BKW algorithm is able to solve the LPN problem with modified data distribution $\mathcal{D}_{CT}$ in Coin-Tossing model.

**Lemma 4.** *(Blum et al. (2003)) Assuming $A$ is a $n \times k$ Boolean matrix generated under distribution $\mathcal{D}$ where $k = 2^{\mathcal{O}(n/\log n)}$, as well as an $k$-bit vector $\hat{y}$ produced by multiplying $A$ by an (unknown) $k$-bit message $x$, and then corrupting each bit of the resulting bitstring $y = xA$ with probability $\eta < 1/2$. The BKW algorithm can recovery the $x$ with (time, data, storage) complexity $poly(k)$ so long as the noise is random and there is no other $y'$ within Hamming distance $o(k)$ from the true $y$.*

---

**Algorithm 1** BKW algorithm

---

**Input:** The $\text{LPN}_{n,\tau}$ Oracle
**Output:** The first bit of the secret $\boldsymbol{s}$

  1: Choose a small number $\varepsilon > 0$
  2: $c := (1 - \varepsilon) \log\!\left(\frac{n}{\tau}\right)$                      $\triangleright$ Set the number of Gaussian-elimination rounds
  3: $d := \frac{k}{c}$                                                  $\triangleright$ Set the block length
  4: $N := \left(c - 1 + \frac{\log^2 n\tau}{(1 - 2\tau)^2} + \log^2 n\right)2^d$             $\triangleright$ Set the query number
  5: Obtain $(A, b) \leftarrow \text{LPN}_{n,\tau}^N,\ A \in \{0,1\}^{N \times n}$
  6: **for** $i = 1, ..., c - 1$ **do**
  7:     **for all** $j \in \mathbb{F}_2^d$ **do**
  8:         **if** A row $\boldsymbol{a}_k$ of $A$ has a suffix $j \,||\, 0^{(i-1)d}$ **then**
  9:             Add $\boldsymbol{a}_k$ to all the other rows with the same suffix $j \,||\, 0^{(i-1)d}$;
10:             Add $b_k$ to the label of the corresponding rows;
11:             Remove the $k$-th row in $(A, b)$.
12:         **end if**
13:     **end for**
14: **end for**

---

The gap for applying the original BKW algorithm to the data generated under CoTo model (denoted as distribution $\mathcal{D}$ in the context) is that we need to prove under the distribution, there are no valid $y'$ such that $d(y, y') = o(k)$. To clarify this, we can using the following trick to avoid the details in the BKW algorithm. Because the algorithm is correct in $k$ random samples, it keeps no other $y'$ within Hamming distance $o(k)$ from the true $y$. For distribution $\mathcal{D}$, the oracle generated $k/2$ sample randomly and $k/2$ samples with encryption primitive. Regardless of the data distribution generated by the latter, the random samples from the former guarantee that no $y'$ exists with a distance to $y$ of $o(k/2) = o(k)$. Then in the coding-theoretic view, this corresponds to producing a $1 - o(1)$ fraction of the desired codeword (i.e. $y$). Although we do not know the remained bits, we can recover the codeword so long as no other codeword is within distance $o(1)$. As for the well-order property of BKW algorithm can be naturally extended from the idea of the algorithm itself since the expected occurrence of noise is less than real. Similarly, the property also can be extended to LCPN problem (under the Coin-tossing distribution). Thus we only need to consider the value of noise rate $\tau$.

Note that the point also can be proved by the method introduced in Feldman et al. (2006) from the agnostic model and construct a new reduction from adversarial noise (considered the conditional noise is controlled by adversary) to random noise.

# B   Omitted Experiment and Details

## B.1   Detect one CPF-concept in Real Machine Learning Algorithm

In the preceding discussion, we examined the complexity of using machine learning (ML) algorithms to learn the CPF-based concept in a CoTo model. However, unlike the previous subsection, we cannot draw rigorous theoretical conclusions due to their complexity and limited interpretability. Nonetheless, it is evident that machine learning algorithms cannot reliably detect all complex feature functions from limited sample sets, even when the corresponding discrepancy $\varepsilon_\theta$ is substantial.

To illustrate this point, consider a differential attack scenario where each input is of the form $\boldsymbol{y}^i = (\boldsymbol{y}_0, \boldsymbol{y}_1) = (\text{Hash}(\boldsymbol{x}_i), \text{Hash}(\boldsymbol{x}_i \oplus a))$, with $\text{Hash}(\cdot)$ being a cryptographically secure hash function and $a$ is a constant. There exists a concept $\theta(\boldsymbol{y}) = \text{Hash}^{-1}(\boldsymbol{y}_0) \oplus \text{Hash}^{-1}(\boldsymbol{y}_1 \oplus a)$ with a high discrepancy. Nevertheless, utilizing modern deep learning algorithms to fit this function in limited data and time complexity is nearly impossible due to the inherent difficulty in inverting hash functions within well-established cryptographic primitives unless $\mathcal{P} = \mathcal{NP}$, see Shalev-Shwartz & Ben-David (2014).

It highlights a limitation of neural distinguishers. Conversely, there are many machine learning algorithms, each of which may perform differently depending on the specific task. Intuitively, we can explore which Boolean functions are learnable by particular learning algorithms and determine

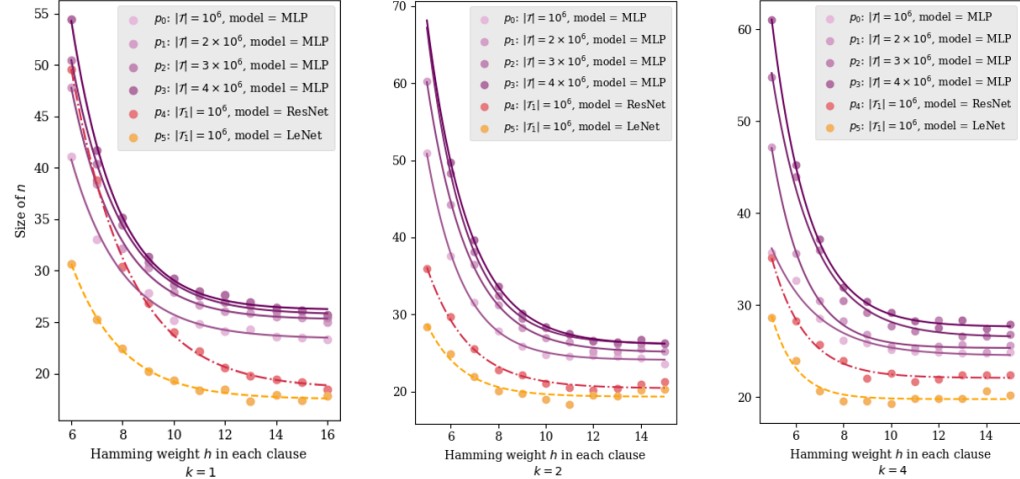

Figure 2: The results for learning algorithm (MLP, ResNet, LeNet) leveraged in Coin-Tossing model with a CPF concept $\theta$ for $k = 1, 2, 4$, $\varepsilon_\theta = 0.25$, $\lambda = 0.1$.

Each data point represents an average result from repeated trials for a specific Hamming weight and fixed dataset size. The curves in the figure represent the fitted results using an exponential decay distribution. For any point $(h, n)$ where $h \geq 1$ and $n \geq h$, and its corresponding parameters, if it lies below the curve, the learning algorithm is likely to succeed; otherwise, it is less likely.

whether they belong to the set $\Theta_\lambda$. If the two sets intersect, it can be concluded that the primitive is not learning-resistant.

However, research in this area remains limited, making it challenging to generalize findings across different algorithms. Moreover, for most symmetric primitives and typical cryptanalysis scenarios, the complete set $\Theta_\lambda$ is either unknown or impractically large. As a result, a more pragmatic direction is to explore how intrinsic properties of Boolean functions relate to the effectiveness of machine learning models. While this approach lacks rigorous proof and may not be universally applicable to all Boolean functions or algorithms, it provides valuable insights into whether machine learning techniques can effectively capture certain families of Boolean functions.

Now, our objective is to identify indicators that suggest whether a Boolean function can be effectively learned. The previous theoretical results suggest that key parameters influencing the complexity of the problems include $n$, $k$, and the Hamming weight $h$ of each clause in a CPF concept. Thus, it is nature to think these parameters will also affect the complexity of ML model learning these concepts. To demonstrate it mathematically, we introduce a implicit function $\Phi$ to assess the complexity of a machine learning algorithm $\mathcal{A}$ to learn a CPF concept $\theta$ using training data $\mathcal{T}$ from the CoTo model:

$$\Phi_{\mathcal{A}}(\theta) = \phi_{\mathcal{A}, \mathcal{T}}(n, k, h) \tag{5}$$

Here, for a given $\theta$, $\mathcal{T}$ and $\mathcal{A}$, $\Phi_{\mathcal{A}}$ is represented by the function $\phi$, which is defined over three hidden variables $n$, $k$, and the clause Hamming weight $h$ of the concept $\theta$. The following experiment shows that the $\phi$ is subject to exponential decay with respect to $n$ and $h$.

**Experiment** To simulate the encryption primitive in the CoTo model, we can use a generative dataset with a fixed discrepancy (we take 0.25 here) for a given concept $\theta$, where each $s_j$ is chosen at random with a predefined Hamming weight $h$. We evaluate several learning algorithms, including MLP, ResNet, LeNet (LeCun et al. (1998)), and Transformer (Vaswani et al. (2023)) For each $h$, we vary the input size $n$ to identify, via binary search, an approximate upper bound on $n$ at which the learned function $\hat{\theta} \in \Theta_\lambda$ for $\lambda = 0.1$. For each Hamming weight, the search is repeated 32 times. Because the position of the secret had a slight effect on the performance of the learning algorithm, repeated experiments were used to remove fluctuations. We record the resulting mean values in Figure 2 and fit them using an exponential decay model. Throughout the experiment, we set $k$ to 1, 2, and 4.

Figure 3: The round encryption function in SPECK algorithm (Jean, 2016).

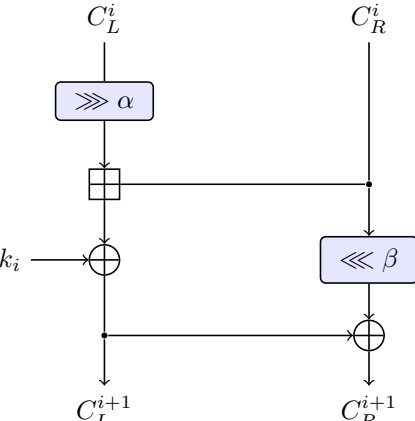

The MLP model consists of two fully connected layers with 128 and 64 neurons, respectively, using rectified linear unit (ReLU) activation functions. The second layer connects to a prediction head with a Sigmoid activation to output binary probabilities. The LeNet model includes three convolutional layers, each followed by an average pooling layer, and two fully connected layers with 120 and 84 neurons, respectively, all using ReLU activations. The final layer is a Sigmoid-activated prediction head. For the ResNet model, we adopt Gohr's neural network architecture, composed of three residual blocks followed by two fully connected layers (with 128 and 64 neurons, respectively). The experiments were conducted on 4 Tesla T4(15GB) GPUs, utilizing a total of 240 GPU days.

**Results and Conclusions** As shown in Figure 2, when evaluating a single concept under the Coin-Tossing model, the MLP model generally outperforms both ResNet and LeNet, although all three follow a similar exponential decay trend. For a fixed training dataset size and clause number $k$, the complexity curve for each model closely fits an exponential decay function. Holding other parameters constant, we observe that increases in the input size $n$ and the clause Hamming weight $h$ result in increased learning difficulty. Additionally, when $h$ is small, variations in $k$ significantly affect performance. These findings, combined with earlier complexity results, highlight a key insight: simplifying the representation of the target Boolean concept improves the effectiveness of neural distinguishers.

### B.2 DETAILS IN BUILDING NEURAL DISTINGUISHERS

The state in SPECK32/64 consists of two 16-bit words $(C_L, C_R)$. In the $i$-th round, the encryption algorithm will compute the next state with:

$$C_L^{i+1} := ((C_L^i \ggg 7) \boxplus C_R^i) \oplus k_i, C_R^{i+1} := (C_R^i \lll 2) \oplus C_L^{i+1}$$

Here, $k_i$ denotes the $i$-th subkey, derived from the 64-bit master key by performing the same round function. The notation $\ggg$ and $\lll$ represent right and left bitwise rotations, respectively, and $\boxplus$ denotes addition modulo $2^{16}$. The Figure 3 shows the round function in SPECK32/64.

Gohr (2019) introduced a method for generating a positive sample set by setting the input generating functions $I_0 = x_i$ and $I_1 = x_i \oplus \Delta$, with $\Delta = (0x0040, 0x0000)$ and $x_i$ are generated randomly. The output function reshapes the encrypted data into $4 \times 16$ tensors. This setup was used to construct datasets within the CoTo model, with training sets sized at $10^8$. Using this approach, Gohr successfully trained ResNet-based distinguishers for 5, 6, and 7 rounds of SPECK32/64.

However, training an 8-round distinguisher directly was impractical, so they employed a retraining-based method instead:

1. Retrain the best 7-round distinguisher to recognize 5-round SPECK32/64 with the input difference $(0x8000, 0x840a)$, which is the 3-round input difference transfer to the output difference $(0x0040, 0000)$ with high probability.

2. Train a new distinguisher using $(0x0040, 0x0000)$ as the fixed input difference.

In step 2, they use $2 \times 10^9 \approx 2^{30.897}$ encryption pairs to train their model, which requires a prolonged training period due to the high data complexity. Moreover, constructing a 9-round neural distinguisher seems impractical using either direct training or retraining. The chosen input difference $(0x0040, 0x0000)$ transitions deterministically into a low Hamming weight difference, and prior work Benamira et al. (2021) showed this difference also yields the highest differential probability for 3- or 4-round SPECK32/64. Based on these observations, they concluded that using the highest-probability difference for $r - 1$ or $r - 2$ rounds is effective for training a distinguisher on $r$ rounds.

**Building 8-round and 9-round Neural Distinguishers for SPECK32/64** Building on the analysis in Sections 5, we propose a new method to train neural distinguishers for 8 and 9 rounds without modifying the neural network architecture. Let $E_{r\text{-Speck}}$ denote the $r$-round of SPECK32/64. Again, for a fixed primitive and learning algorithm, the hypothesis space and characteristic function class $\Theta_\lambda$ depend on the generating functions. Therefore, choosing these generating functions is an important issue. Gohr introduced a greedy search to find input differences, while Hou et al. (2021) proposed a SAT-based search, and Bellini et al. (2023b) introduced a metric-based approach using bias scores. However, these approaches struggle to handle high Hamming weight input differences or deep rounds.

By contrast, our results also show that input differences with high Hamming weight can also train better neural network distinguishers. From both the perspective of problem complexities and prior experimental results in Section 3, the size of $n$ has an exponential impact on the difficulty of learning features. Consequently, our first strategy is to reduce $n$ by modifying the output generating function, i.e., the *compression* technique. As previously mentioned, the concepts in 8-round and 9-round are likely more complex than for fewer rounds in SPECK32/64, which also have a low discrepancy. To address this, we can compress the input vector sample by employing the output generating function $\omega(\boldsymbol{y}^i) = \boldsymbol{y}_0^i \oplus \boldsymbol{y}_1^i$. Then it has $\boldsymbol{y}^i = (E_k(x_0) \oplus E_k(x_0 \oplus \Delta)) \in \mathbb{F}_2^{32}$. Compression may reduce some characteristics, while simplifying others and potentially improving the ability of learning algorithms to capture them.

Specifically, our experiments in the previous section suggest that the Hamming weight $h$ correlates with the complexity of the CPF function learned by some machine learning algorithms, with fixed $k$ and $n$. For example, when $k = 1$, the results shown in Figure 2 illustrate that it is nearly impossible to learn features with $h \geq 12$ at 64 bits (under the settings provided in the experiment). In contrast, for $n = 32$, while the probability of learning concept shows a declining trend with the increasing $h$, there remains a significant difference compared to the 64-bit case.

Theoretically, compression maps the element from $\mathbb{F}_2^{64}$ to $\mathbb{F}_2^{32}$; thus, it reduces the complexity of learning a general CPF characteristic from $2^{\mathcal{O}(n/\log n)}$ to $2^{\mathcal{O}(n/2 \cdot \log(n/2))}$, as shown in Corollary 2 (assuming the discrepancy is constant). Similarly, by applying Theorem 3, it can reduce the complexity of learning a certain output differential from $\text{poly}(n^2, \cdot, \cdot, \cdot)$ to $\text{poly}((n/2)^2, \cdot, \cdot, \cdot)$. Then the compression makes the $\mathcal{ND}$ able to learn more difficult characteristics. On the other hand, the technique still preserves the differential-based or DL characteristics, although some others, such as the linear characteristics, will be eliminated.

The second idea follows naturally as well if we know a characteristic function $\theta \in \Theta_\lambda$ and $\hat{\varepsilon}_\theta$ under particular input generating functions $\alpha_i(x)$, then there will potentially exist another characteristic function $\theta' \in \Theta_\lambda$ with $\hat{\varepsilon}_{\theta'} \geq \hat{\varepsilon}_\theta$. To find an appropriate difference $\Delta$ which can create one (or multiple) characteristic function(s) with a high discrepancy, we employ the MILP/MIQCP tool proposed in prior work Bellini et al. (2023a). And we choose the input difference $\Delta_1 = (0x2800, 0010)$ for 8-round SPECK32/64. The following steps formalize the new training method:

1. Construct a CoTo model with:
   - Setting input generating functions: $\alpha_0(x_i) = x_i$, $\alpha_1(x_i) = x_i \oplus \Delta_1$.
   - Setting output generating function: $\omega(t_0, t_1) = t_0 \oplus t_1$.
   - Generate a training set $\mathcal{T}$ of size $2^{19}$ and a test set $\mathcal{T}'$ of size $2^{23}$.

2. Train a ResNet model (as in Gohr (2019)) on $\mathcal{T}$ for up to 50 epochs. The resulting model is denoted $\mathcal{ND}_{\text{comp}}^1$, where the subscript refers to the compression-based approach. The other hyperparameters, e.g., learning rate, are kept the same with Gohr (2019).

It is worth noting that we did not first use the difference $(0x2800, 0x0010)$ for training. This difference has been discussed in several works (Gohr (2019); Benamira et al. (2021)). Specifically, the existed result in but in fact it is only in this paper that it is shown to be superior to the difference $(0x0040, 0x0000)$.

Furthermore, we also try to construct a 9-round SPECK32/64. Bellini et al. (2023a) introduced a 9-round Differential-Linear distinguisher with the characteristic:

$$\Delta_2 = (0xa810, 0x0010) \overset{E_{9\text{-Speck}}}{\longrightarrow} \Gamma = (0x0205, 0x0204)$$

which holds with a practical correlation of $2^{-7.3}$. However, as demonstrated in Experiment 1, this concept is exceedingly difficult to learn because it involves a 64-bit feature search. Similarly, we use the following steps to construct the neural distinguisher:

1. Similarly, construct a CoTo model with an oracle generating positive samples $\mathcal{P}_i = (E_{k_i}(x_i) \oplus E_{k_i}(x_i \oplus \Delta_2))$ and negative samples with probability 0.5. The input generating function are $\alpha_0(x_i) = x_i$, $\alpha_1(x_i) = x_i \oplus \Delta_2$, while the output generating function is $\omega(t_0, t_1) = t_0 \oplus t_1$. Use the oracle to construct the training sample set $\mathcal{T}$ and test sample set $\mathcal{T}'$ such that $|\mathcal{T}| = 2^{30}$ and $|\mathcal{T}'| = 2^{23}$.

2. Train a ResNet model on $\mathcal{T}$ for up to 200 epochs. The resulting distinguisher is denoted $\mathcal{ND}^2_{\text{comp}}$.

The related results are in Table 2. Due to the clear mechanism proposed in Section 3, the training process of the 8-round distinguisher does not require complicated retraining and plenty of data. Moreover, the distinguisher has great improvements in accuracy.

**Benchmark Selection** The increasing use of neural networks as distinguishers in symmetric cryptanalysis necessitates a rigorous and fair benchmarking methodology. A foundational step in establishing such a benchmark is to define the adversary's capabilities. We assume a standard differential cryptanalysis scenario where the adversary has access to a differential oracle. This oracle, given a plaintext $x$ and a chosen input difference $\Delta$, returns the corresponding ciphertext pair $(E_k(x), E_k(x \oplus \Delta))$. For offline data generation, the key $k$ is selected uniformly at random. This setup allows the adversary to freely choose plaintexts and differentials, mirroring a typical differential-cryptanalysis. The works cited in our comparison adhere to this established attack model. Also, the works referenced were the prevailing state-of-the-art results at the time.

While some studies employ multiple differentials to improve the accuracy of neural distinguishers, this approach invariably leads to higher data complexity in practical attack scenarios Benamira et al. (2021). Consequently, to maintain a fair and consistent basis for comparison, researchers often explicitly specify the number of ciphertext pairs utilized by the neural distinguisher.

A potential concern in comparative studies is that varying the input differential could lead to an inequitable assessment of results. However, we argue this concern is unfounded. While extensive research has focused on selecting optimal input differentials to enhance the accuracy of neural network distinguishers Gohr et al. (2022); Bellini et al. (2023b); Hou et al. (2025), our work is the first to identify a differential that surpasses the accuracy of the commonly used (0x0040, 0x0000) as input difference.

In addition to the neural network distinguishers mentioned above, we also use a series of ablation experiments to demonstrate the superiority of our distinguishers, including the same difference experiment without compression (corresponding to the gray row in the table).

**Improved Distinguishers for DES** Equipped with our new theory, it can explain an intriguing issue: in some cases, even if we incorporate key-related information as extended input to a model, it fails to enhance its accuracy, since the algebraic relationship between keys and ciphertexts is still hard to learn. Our framework can also explore some previous related neural distinguishers from a foundational perspective. For instance, in Hou et al. (2020), they constructed a neural distinguisher for DES based on some existing linear characteristics. Their process can be summarized as follows:

1. Generate plaintexts $P$ and master keys $K$ from the random uniform distribution, and encrypt $P$ with $K$ by $n$-round DES and obtain ciphertexts $C$.

2. Given a known linear masks $(\Gamma_P, \Gamma_C, \Gamma_K)$, use $(\Gamma_P \wedge P, \Gamma_C \wedge C)$ as the model input and $\langle \Gamma_K, K \rangle$ as the label.

3. Train a ResNet model on the resulting dataset.

Their best result was based on the following correlation:

$$P[7, 18, 24, 29, 47] \oplus C[7, 18, 24, 29, 47] = K_1[22] \oplus K_3[22].$$

Here, $P[i]$ and $C[i]$ denote specific bit positions in the plaintext and ciphertext, and $K_i$ refers to the $i$-th round subkey in DES. Because they use the right hand of the characteristic as the label, the experiment can be interpreted as learning a parity concept. Moreover, their experiment can be regarded as an example of our experiment in Appendix B.1.

By setting irrelevant bits to zero, they reduced the complexity of finding relevant variables, and the model can approximately fit the characteristic. However, the model fell short of the theoretical accuracy: while the expected accuracy was $0.7$, the achieved accuracy was only $0.668$. A simple yet effective improvement would be to remove the irrelevant bits entirely, rather than zeroing them out, or alternatively to introduce auxiliary bits indicating the relevance of each input position. Namely, we set the output generating function as $\omega(\boldsymbol{y}_0, \boldsymbol{y}_1) := (\pi_{\Gamma_P}(y_0), \pi_{\Gamma_C}(y_1))$ where projection maps $\pi_c : (a_0, a_1, ..., a_{n-1}) \mapsto (a_{c_{i_0}}, a_{c_{i_1}}, ..., a_{c_{i_{n-1}}})$, $c_{i_k}$ denote the index of the $k$-th one in the bitstring $c$.

A similar issue arises in the work by Zahednejad & Lyu (2022), which trained a neural network that can only learn a 3-round SPECK32/64 integral characteristic with an input multiset of 4 active bits in the right word is:

$$(cccccccccccccccc, ccccccccccccaaaa),$$

where $c$ denotes constant bits and $a$ represents active bits. However, their model faced similar challenges and failed to fit the 8-round integral characteristic due to the same fundamental limitation. Incorporating with our method, the accuracy can achieve the idea accuracy.

**Selections of Output Generating Functions**  Within this study, we propose two distinct output generating functions to augment the problem-solving capabilities of deep-learning model.

1. The compression function: $\omega(\boldsymbol{y}_0, \boldsymbol{y}_1, ..., \boldsymbol{y}_m) = \boldsymbol{y}_0 \oplus \boldsymbol{y}_1 \oplus ... \oplus \boldsymbol{y}_m$.

2. The projection mapping: $\omega(\boldsymbol{y}_0, \boldsymbol{y}_1, ..., \boldsymbol{y}_m) = (\pi_{c_0}(\boldsymbol{y}_0), \pi_{c_1}(\boldsymbol{y}_1)), ..., \pi_{c_m}(\boldsymbol{y}_m))$ for some constant $c_i \in \mathbb{F}_2^l$.

These functions are characterized by their shared ability to preserve designated concepts after application, while markedly reducing the upper bound on the complexity associated with learning these concepts in our model. A further effect of these functions is the elimination of certain latent concepts from the original concept class. The task of selecting suitable output generating functions may thus be viewed as a trade-off.

To carry out an attack on the SPECK32/64 cipher, we first utilize state-of-the-art MILP/MIQCP methodologies to ascertain a known concept within the set $\Theta_\lambda$ where said concept adheres to the formulation of a Differential-Linear (DL) characteristic. Subsequently, our analysis is predicated on the assumption that $\Theta_\lambda$ is richly populated with other concepts that also satisfy the definition of a DL characteristic. Recall the DL characteristics have the form of $\langle \Gamma, \boldsymbol{y}_0 \rangle \oplus \langle \Gamma, \boldsymbol{y}_1 \rangle$. Then performing the compression function will turn the concept it into $\langle \Gamma, \omega(\boldsymbol{y}_0, \boldsymbol{y}_1) \rangle$ which will keep the concepts in $\Theta_\lambda$.

This demonstrates the flexibility in selecting the output generating function, for which our theory provides a core criterion. This work opens up an avenue for future research, where more elaborate output generating functions can be developed based on our proposed principle. Such functions could be enhanced by integrating more domain-specific prior knowledge.

## C  THE USAGE OF LARGE LANGUAGE MODELS

The authors only used the Large Language Models to polish our manuscript and find typos. The authors carefully reviewed all text to ensure accuracy and consistency with our findings and take full responsibility for the entirety of the work.

