# OpenReview forum: "Rethinking Learning-based Symmetric Cryptanalysis: a Theoretical Perspective"
_ICLR.cc/2026/Conference — ICLR 2026 Conference Withdrawn Submission_

### Official Review · Reviewer_JatA · 2025-10-29

**Soundness:** 3
**Presentation:** 3
**Contribution:** 2
**Rating:** 2
**Confidence:** 2

**Summary:**

This paper proposes a theoretical framework for explaining the performance of neural distinguishers, consisting of the Coin-Tossing (CoTo) model and the Conjunctive Parity Form (CPF). The CoTo model formalizes the data generation process, while the CPF provides a unified representation of conventional distinguishers. The authors prove that any concept belonging to the CPF class is learnable in sub-exponential time. Finally, based on this result, they construct a neural distinguisher for SPECK32/64, extending existing neural attacks from 8 to 9 rounds for the first time.

**Strengths:**

- The paper introduces the CPF, which unifies a wide range of existing distinguishers, and theoretically proves its learnability.
- It improves the performance of neural distinguishers on SPECK32/64.

**Weaknesses:**

- The approach might ultimately only cover the scope of existing conventional distinguishers.
- For the 9-round results, the paper does not include a comparison that incorporates the differential-linear results used to determine the input generation function, so the effectiveness of the proposed method remains unclear.
- Theoretical results only demonstrate the existence of a learning algorithm, without providing any concrete construction or procedure.

**Questions:**

## The neural distinguisher might still lie within the scope of conventional distinguishers.

Expressing conventional distinguishers in the unified form of CPF and proving their learnability has theoretical value. However, because the CPF framework is designed to include conventional distinguishers, it raises the concern that the neural distinguisher may simply be re-learning what classical distinguishers already capture. For example, Benamira et al. demonstrated that results similar to Gohr's neural distinguisher can be achieved through traditional cryptanalytic methods. In Example 1 and Example 2, the paper shows that CPF concepts can represent classical differential and differential-linear distinguishers, but this also implies that the neural network may merely be learning these existing structures. A particularly concerning point is that, in order to generate training data for the neural distinguisher, one still needs to rely on conventional cryptanalysis to design appropriate input generating functions and output generating functions.
Ideally, these should also be learned automatically, but since they are determined using classical analysis, the neural distinguisher may in fact just be imitating conventional distinguishers. If that is the case, the necessity of employing machine learning at all becomes questionable.

## The effectiveness of the 9-round neural distinguisher on SPECK32/64 is unclear because the corresponding differential-linear results are not reported.

The paper lists the extension of the neural distinguisher to 9 rounds as one of its main contributions. However, to estimate the input difference $\Delta$, the authors used the MILP/MIQCP-based method proposed by Bellini et al. (2023a). During this process, the corresponding $\Gamma$ is also computed, meaning that a differential-linear distinguisher using this $(\Delta, \Gamma)$ pair can be obtained. Therefore, Table 2 should include a comparison of the accuracy, TPR, and TNR achieved by this differential-linear distinguisher. If its results are almost the same as those of the proposed neural distinguisher, then, as noted in Concern 1, the neural approach would not go beyond the scope of existing distinguishers, and its advantage would be unclear. Moreover, the claimed contribution of extending the distinguisher to 9 rounds might simply reflect the reuse of a conventional differential-linear distinguisher rather than a genuinely new capability of the neural model.

## Theorem 3 only proves the existence of a learning algorithm without showing how to construct it.

This limitation may be inherent to the PAC-learning framework, but the paper only establishes the existence of a learning algorithm, without specifying how such an algorithm could be concretely constructed. From a cryptanalytic perspective, researchers are also interested in how a distinguisher can be explicitly built. Providing even a brief description or outline of how the learning algorithm might be realized would considerably strengthen the paper's practical value.

---

### Official Review · Reviewer_v52A · 2025-10-30

**Soundness:** 3
**Presentation:** 2
**Contribution:** 2
**Rating:** 4
**Confidence:** 3

**Summary:**

The paper presents a theoretical framework for analyzing learning-based cryptanalysis of symmetric primitives, particularly SPNs. It defines a family of distinguishers $\Theta_\lambda$ through an advantage constraint and introduces concepts to formalize the success of learning-based adversaries independently of specific neural architectures. The work focuses on conceptual unification. The application section illustrates how the proposed theoretical framework can guide the design of neural distinguishers for practical ciphers such as SPECK32/64. While the reported accuracy improvements are modest, these experiments serve to demonstrate the potential applicability of the Coin-Tossing model and CPF formalism to real-world neural cryptanalysis.

**Strengths:**

Its main strength is the theoretical unification and conceptual rigor it brings to an area that has mostly been explored empirically.

**Weaknesses:**

While the paper presents a rigorous theoretical model of learning-based cryptanalysis, its focus appears primarily cryptographic rather than machine-learning-oriented. The contribution centers on formalizing a generalized attack framework for SPNs and analyzing properties of distinguishers, but it remains unclear how these abstractions translate into improved or novel neural-network–based attacks in practice. The authors may wish to demonstrate how the proposed generalization informs or enhances current NN-based cryptanalytic methods.

**Questions:**

Q1: The definition of the function class $\Theta_\lambda$ is somewhat ambiguous and appears to deviate from standard set-roster notation. It is unclear whether it is meant to denote (i) a specific model parameterized by $\lambda$, or (ii) the set of all functions realizable under the constraint (1). If the latter is intended, the notation would benefit from an explicit clarification. Making this precise would also make Remark 1 (closure under output flipping) more transparent.

Q2: There is some ambiguity in the notation for $\mathcal{Y}$ and $\mathcal{R}$. Earlier, these symbols denote finite subsets of the training dataset (structured and random samples, respectively), but later they appear to refer to the underlying distributions from which the samples are drawn, as said after Eq. (1): "Here, y and r are randomly drawn from $\mathcal{Y}$ and $\mathcal{R}$, respectively" (line 188).

Minor remark: in the definition of Oracle_CT in line 197 the operator I_1 appears unnecessarily.

---

### Official Review · Reviewer_yb2Y · 2025-11-01

**Soundness:** 3
**Presentation:** 3
**Contribution:** 2
**Rating:** 6
**Confidence:** 2

**Summary:**

This paper provides a theoretical framework for symmetric cryptanalysis.
* The authors introduce the Coin-Tossing (CoTo) model to formalize the construction of distinguishers and also propose the Conjunctive Parity Form (CPF) as a unified algebraic representation for distinguishers.
* They provide a proof that any CPF concept is learnable in sub-exponential time and even in polynomial-complexity time in certain cases.
* The theoretical findings lead to a practical improvement in neural distinguishers for a standard block cipher by reducing the problem’s complexity upper bound.

**Strengths:**

* This theoretical framework is novel with respect to cryptanalysis, addressing a gap in the literature.
* The complexity results are interesting and directly lead to practical impact for designing better neural distinguishers, as demonstrated by the results on SPECK 32/64.
* The definitions, theorems, and proofs are well-presented. The authors also provide enough background for readers who may not be familiar with the field.

**Weaknesses:**

* Other cryptanalysis works have shown that there can be a gap between theory and practice (e.g. Benchmarking Attacks on Learning with Errors by Wenger et al.). In this case, the work establishes a theoretical upper bound, but it seems unclear as to whether this reflects actual neural network performance.
* The compression technique reduces problem complexity but also would eliminate some features. Can you discuss in more detail the trade-offs of this technique and in which cases it could affect distinguisher performance?
* Some minor typos (e.g. missing space in line 28 and “Boolen” in line 234)

**Questions:**

How would the practical results generalize to other ciphers with different structural properties? Are there certain cipher structures where this technique would perform better (or worse)?

---

### Official Review · Reviewer_3K9D · 2025-11-01

**Soundness:** 2
**Presentation:** 3
**Contribution:** 2
**Rating:** 2
**Confidence:** 3

**Summary:**

Authors propose a general Coin-tossing model that connects the LPN problem to cryptanalysis learning tasks. The model operates at a high level of abstraction, which ensures it covers a variety of cryptanalysis schemes. The complexity of the coin-tossing model is then explored and as a result, authors propose using a different output-generating function that improves the accuracy for 8 and 9 round Speck over Gohr's results.

**Strengths:**

- The generality of the coin-tossing model to unify various cryptanalysis approaches is nice.
- The connection to the LPN problem and the analysis of learning complexities are interesting.
- The accuracy improvement on the 8-round Speck is impressive.

**Weaknesses:**

- Varying output generating functions is not novel. This has been explored by several other works, including precisely the proposed compression function (the Multi-output Difference in [1]). Several more approaches have been studied; this is referred to as feature engineering in this survey[2].
- While the coin tossing model is very general, the experimental sections only test two ciphers in very narrow scenarios. Evaluating whether different output-generating functions also help in other cases (i.e., Simon, which is a common target across the literature) would help in evaluating the generality of insights.
- The performance improvement in the 8/9 round Speck case is impressive, but neural distinguishers with higher accuracies have been found by using more ciphertexts (or their compressed version[1], see the tables in[2]).

References:
[1]:  Hou, ZeZhou, Ren, JiongJiong, Chen, ShaoZhen, Improve Neural Distinguishers of SIMON and SPECK, Security and Communication Networks, 2021, 9288229, 11 pages, 2021. https://doi.org/10.1155/2021/9288229

[2]: Gerault, D., Hambitzer, A., Huppert, M., & Picek, S. (2024). SoK: 6 Years of Neural Differential Cryptanalysis. Cryptology ePrint Archive.

**Questions:**

- W2: As neural distinguishers with m>2 can perform better than those with m=2 in some cases [2], would the output generating functions also help in those cases?
- While using different output-generating functions is motivated by the theoretical results, the generality of reducing input size n being better for complexity seems dependent on the specific cipher being analyzed. Does the coin-tossing model help us construct these output-generating functions in a more principled manner?
- The improved performance in Table 2 over Gohr(2019) has much more imbalanced TPR vs TNR. Does this affect key recovery?
- Do models trained with compression function require less training data/epochs vs models trained on the standard output differences?
- What was the performance of ND^1_comp with the original output difference from Gohr?


Minor/Typos:
- I_0 and I_1 are used in appendix B.2 as input generating functions over \alpha_0 and \alpha_1. (there is also an I_1 left in Definition 2)
- Page 2 The learning theory and LPN problem -> Learning Theory and the LPN problem.
- Table 2: Nerual Distinguisher
- L899: nature->natural
- L912: "and Transformer" there are no Transformers in the figure, also the citation should maybe be for the published version if included.
- L1028: unclear sentence



References:
[1]:  Hou, ZeZhou, Ren, JiongJiong, Chen, ShaoZhen, Improve Neural Distinguishers of SIMON and SPECK, Security and Communication Networks, 2021, 9288229, 11 pages, 2021. https://doi.org/10.1155/2021/9288229

[2]: Gerault, D., Hambitzer, A., Huppert, M., & Picek, S. (2024). SoK: 6 Years of Neural Differential Cryptanalysis. Cryptology ePrint Archive.

---

### Note · Authors · 2025-11-20

**Comment:**

We would like to sincerely thank the anonymous reviewers for the time and effort invested in evaluating our manuscript. Although we do not fully agree with some of the reviewers’ assessments of the weaknesses of our work, we acknowledge that the current version of the paper still has room for improvement. We hope to submit an improved version of this work to an appropriate venue in the future.

Thank you again for your time and consideration.

**Withdrawal Confirmation:**

I have read and agree with the venue's withdrawal policy on behalf of myself and my co-authors.